# Temporary Employment Is Associated with Poor Dietary Quality in Middle-Aged Workers in Korea: A Nationwide Study Based on the Korean Healthy Eating Index, 2013–2021

**DOI:** 10.3390/nu16101482

**Published:** 2024-05-14

**Authors:** Seong-Uk Baek, Myeong-Hun Lim, Yu-Min Lee, Jong-Uk Won, Jin-Ha Yoon

**Affiliations:** 1Department of Occupational and Environmental Medicine, Severance Hospital, Yonsei University College of Medicine, Seoul 03722, Republic of Korea; 2The Institute for Occupational Health, Yonsei University College of Medicine, Seoul 03722, Republic of Korea; 3Graduate School, Yonsei University College of Medicine, Seoul 03722, Republic of Korea; 4Graduate School of Public Health, Yonsei University College of Medicine, Seoul 03722, Republic of Korea; 5Department of Preventive Medicine, Yonsei University College of Medicine, Seoul 03722, Republic of Korea

**Keywords:** diet, eating habit, health behavior, lifestyle, non-standard employment, precarious employment

## Abstract

Temporary employment is associated with an elevated risk of cardiovascular diseases and mortality. This study explored the association between temporary employment and dietary quality in middle-aged workers. This cross-sectional study included a nationwide sample of middle-aged Korean workers (n = 6467). Employment type was categorized into regular, fixed-term, and daily employment, based on labor contract duration. Dietary quality was assessed using the Korean Health Eating Index (KHEI), which ranges from 0 to 100, with higher scores indicating superior dietary quality. Linear regression was used to estimate beta coefficients (*β*) and 95% confidence intervals (CI). The survey-weighted proportion of regular, fixed-term, and daily employment was 79.0%, 14.2%, and 6.8%, respectively. Fixed-term and daily employment were associated with a reduced KHEI compared with regular employment (*β* [95% CI]: −1.07 [−2.11, −0.04] for fixed-term and −2.46 [−3.89, −1.03] for daily employment). In sex-stratified analysis, the association between temporary employment and dietary quality was more pronounced in men (*β* [95% CI]: −1.69 [−3.71, 0.33] for fixed-term and −2.60 [−4.63, −0.53] for daily employment than in women. In conclusion, this study suggests that temporary employment is a social determinant of dietary quality in middle-aged workers.

## 1. Introduction

In recent years, the trend of globalization and advent of the Fourth Industrial Revolution have brought great changes to employment conditions in the workforce [1]. For instance, the global labor market has witnessed a weakening of the standard employment relationship, which is characterized by a guaranteed retirement age, coverage of social benefits and insurance, and workers’ rights and protection [2]. Instead, temporary employment is increasingly preferred with the informalization of the labor market, marked by short-term contractual arrangements, often based on tasks or a daily basis, in contrast to regular employment with a longer period of contractual labor arrangements [3].

Temporary employment is associated with several adverse health consequences [4]. For instance, previous studies have shown that temporary employment is associated with an increased risk of cardiovascular disease [5,6] and overall mortality [7,8,9]. Additionally, temporary employment is associated with poor mental health conditions [10]. In the current literature, increased job stress and insecurity induced by temporary employment are assumed to be associated with temporary employment and adverse health outcomes [11]. Therefore, the negative impact of temporary employment and job insecurity on health can be particularly more pronounced among middle-aged workers, who bear a significant burden of family responsibilities [12].

Along with increased job stress, unfavorable health behaviors in temporary employees can serve as an important mechanism leading to cardiovascular diseases and mortality in middle-aged workers [7]. Previous studies have shown that temporary employees are more likely to engage in poor lifestyle behaviors, including smoking, alcohol use, and physical inactivity [13,14]. While poor dietary quality is a major contributor to cardiovascular risk factors, the existing literature has limited evidence regarding the relationship between temporary employment and dietary habits. A Spanish study found that temporary employment is associated with a lower Spanish Healthy Eating Index [15]. A Korean study showed that those with temporary employment were more likely to have unfavorable dietary habits, such as skipping meals and eating alone [16]. Various complex mechanisms can influence the relationship between employment type and workers’ dietary habits; previous studies have shown that temporary employees are more often subject to factors such as high job stress, long working hours, and lower wages [17,18], which can lead to poorer dietary quality.

The Korean Healthy Eating Index (KHEI), based on Korean dietary guidelines, was developed to assess the overall dietary quality, reflecting regional dietary practices and culture. Previous studies have shown that low KHEI scores are associated with metabolic diseases [19] and type 2 diabetes [20]. However, to the best of our knowledge, the association between temporary employment and the KHEI has not yet been explored. Furthermore, previous studies have suggested that the association between temporary employment and lifestyle varies depending on sex. For instance, considering middle-aged men’s traditional role as breadwinners, temporary employment may have a greater impact on perceived insecurity and stress among male workers, potentially leading to various effects on lifestyle behaviors based on sex [12,14]. However, differences in sex in the association between temporary employment and dietary quality have not yet been examined.

Therefore, utilizing a nationally representative sample of workers, this study aimed to investigate the relationship between temporary employment and dietary quality among middle-aged workers, while examining potential differences in sex in this association.

## 2. Materials and Methods

### 2.1. Study Sample

This study utilized the raw data of the nutritional survey of the Korea National Health and Nutritional Examination Survey (KNHANES) [21]. The KNHANES is an ongoing nationwide survey that includes a nationally representative sample of people living in Korea. To select a representative sample, the KNHANES employed a multistage probability sampling design, wherein administrative regions in Korea were considered primary sampling units, and households in each region were considered secondary sampling units [21]. As information on the KHEI has been collected since 2013, we included survey participants in the KNHANES from 2013 to 2021. The KNHANES response rates ranged from 75% to 82% during the survey period [22]. Raw data from the KNHANES are available at https://knhanes.kdca.go.kr/knhanes (accessed on 23 December 2023) [21]. Survey weights were allocated to individuals to improve the generalizability of the survey sample and mitigate bias resulting from non-responses.

A flowchart illustrating the sample selection process is shown in Figure 1. Initially, 45,957 adult participants were surveyed during the 2013–2021 survey. Subsequently, we limited our survey sample based on the following criteria: (i) wage workers; (ii) middle-aged workers (aged 40–60 years); (iii) absence of the following chronic diseases: dyslipidemia, diabetes, or hypertension; and (iv) observations without missing values. After applying these criteria, 6467 middle-aged workers were included in the final sample.

Given that this study involves secondary data analysis, we did not compute a sample size for analysis, as is customary with studies that use KNHANES datasets [23,24,25,26,27]. The KNHANES dataset is constructed to encompass a nationally representative sample of the Korean population, which corresponds to our intended study population.

### 2.2. Ethics Statement

All the survey participants provided informed consent. Before conducting each wave of the KNHANES, ethical approval was obtained from the Institutional Review Board (IRB) of the Korea Disease Control and Prevention Agency (approval numbers: 2013-07CON-03-4C; 2013-12EXP-03-5C; 2018-01-03-P-A; 2018-01-03-C-A; 2018-01-03-2C-A; 2018-01-03-5C-A). Additionally, this secondary data analysis was approved with exempt status by the IRB of Yonsei Health System (IRB number 4-2023-0959) prior to conducting the analysis. The authors assert that all procedures contributing to this work comply with the ethical standards of the relevant national and institutional committees on human experimentation and with the Helsinki Declaration of 1975, as revised in 2013. 

### 2.3. Variables

#### 2.3.1. Temporary Employment

Based on the official classification system of Statistics Korea, employment types were categorized according to the duration of the employment contract (regular, fixed-term, and daily employment). Regular employment guarantees at least one year or until retirement; fixed-term employment lasts for more than one month but less than a year; and daily employment encompasses contracts lasting less than a month or daily. Statistics Korea merges permanent employment and long-term temporary employment with contracts lasting one year or longer into a unified classification of regular employment. This is because any employment contract exceeding one year entails entitlements equivalent to retirement benefits, bonuses, and social provisions as permanent employment according to Korean labor law [28]. This categorization has been employed in prior research to examine the health implications of temporary employment [29,30,31].

#### 2.3.2. KHEI

The dietary quality of the survey participants was based on the KHEI developed by the Korean Disease Prevention and Control Agency [32]. The KHEI, comprising 14 items, encompasses adequacy, moderation, and balance components. Adequacy, with eight items, evaluates the sufficient consumption of recommended foods, such as mixed grains, fruits, vegetables, and dairy products. Moderation, represented by three items, evaluates the controlled intake of saturated fatty acids, sweets or sweetened beverages, and sodium. The balance component, with three items, assesses the balance of total energy, carbohydrates, and fat intake. Detailed information on the scoring system of the KHEI was presented in a previous study [32]. The total KHEI score ranges from 0 to 100, with a higher score indicating superior dietary quality and adherence to the Korean dietary guidelines. In the KNHANES, information on dietary intake was collected using the 24-h recall method, and total energy and nutrient intakes were computed using the Standard Food Composition Table, eighth version [33]. All dietary information was gathered through face-to-face interviews conducted by trained dieticians [32]. 

#### 2.3.3. Covariates

The following covariates were considered confounders in our analysis. Sex (men or women) was adjusted for. Age was categorized into 40–44, 45–49, 50–54, and 55–60 years. Educational level was categorized as college or higher, high school, or middle school or lower. The income level was categorized based on the quartile values of monthly income for each year (Q1, Q2, Q3, and Q4). Marital status was categorized as married, unmarried, and others (divorced, widowed, or separated). Occupation type was categorized as white collar, service or sales worker, or blue collar, based on the Korean Standard Classification of Occupations. Weekly working hours were categorized into <35 h, 35–40 h, and ≥53 h based on the working hour policy in Korea [34]. 

### 2.4. Statistical Analysis 

For the descriptive analysis, we first explored the basic characteristics of the survey participants according to employment type. Subsequently, we explored the characteristics of the dietary intake of survey participants according to employment type.

For regression analysis, we explored the association between temporary employment and dietary quality compared to regular employment using crude (Model 1) and adjusted (Model 2) linear regression models. Before fitting the linear regression models, we examined the normal distribution of the KHEI among the survey participants using both a Q-Q plot and a histogram. In Model 3, we tested the moderating effect of sex by including interaction terms between temporary employment and sex. Finally, we stratified our sample by sex and investigated how the relationship between temporary employment and dietary quality manifested differently for each sex. Effect sizes were presented as beta coefficients (*β*) and 95% confidence intervals (CI). R software (version 4.2.3; R Foundation for Statistical Computing, Vienna, Austria) was used for all statistical analyses. For both descriptive and regression analyses, survey weights were adjusted considering the complex survey design of the KNHANES with the R package “survey” and its function “svyglm”.

In our additional analysis, we examined how the relationship between temporary employment and the KHEI differs depending on the employment status of the spouse. We conducted a stratified analysis, comparing groups with a working spouse against those with a non-working spouse or no spouse at all.

## 3. Results

The survey-weighted proportions of regular, fixed-term, and daily employment were 79.0%, 14.2%, and 6.8%, respectively (Table 1). The proportion of those with old age, low educational attainment, low income, and non-white-collar employment was higher among temporary employees (fixed-term or daily employment) than among regular employees. Additionally, the proportion of women and service/sales workers was higher among fixed-term employees, whereas the proportion of men and blue-collar workers was higher among daily employees than among regular employees.

The characteristics of the KHEI among the survey participants are presented in Table 2. The mean KHEI scores were 63.0, 63.2, 63.4, and 60.3 for the overall sample, regular employees, fixed-term employees, and daily employees, respectively. Daily employees scored lower on items related to adequate intake of fruits, vegetables, and dairy products, as well as on the balance component. The Q-Q plot and histogram of the distribution of KHEI among the study sample are presented in Appendix A.

The associations between temporary employment and the KHEI are presented in Table 3. In the fully adjusted model (Model 2), fixed-term employment was associated with a reduced KHEI score compared with regular employment (*β* [95% CI]: −1.07 [−2.11, −0.04]). Additionally, daily employment was associated with a reduced KHEI score compared with regular employment (*β* [95% CI]: −2.46 [−3.89, −1.03]). The *p*-value for the interaction between the male sex and fixed-term employment in the KHEI was 0.086, whereas the *p*-value for the interaction between the male sex and daily employment in the KHEI was 0.020 (Model 3). 

The results of the sex-stratified analyses are presented in Table 4. In male workers, the *β* (95% CI) for the association between fixed-term or daily employment and KHEI was −1.69 (−3.71, 0.33) or −2.60 (−4.63, −0.53), respectively, compared with regular employment. In female workers, the *β* (95% CI) for the association between fixed-term or daily employment and KHEI was −0.90 (−2.04, 0.25) or −1.34 (−3.19, 0.52), respectively, compared with regular employment.

Appendix A presents the results of an additional analysis, showing that among women whose spouse is not employed or who do not have a spouse, temporary employment is associated with lower dietary quality compared to regular employment.

## 4. Discussion

In this study, temporary employment was inversely associated with dietary quality among middle-aged Korean workers. After adjusting for various socioeconomic characteristics, fixed-term and daily employment were associated with reduced KHEI scores compared to regular employment. Additionally, the association between temporary employment and KHEI was more pronounced among male workers than female workers. Therefore, this study highlights that temporary employment may be an important social determinant of dietary quality in middle-aged workers.

Our findings are consistent with those of previous studies showing that temporary employment is associated with unfavorable dietary habits among workers. A similar finding was observed in a cross-sectional study conducted in Spain, where temporary employment showed an inverse association with the Spanish Healthy Eating Index, whereas civil servants and those in permanent employment exhibited a direct association compared to individuals with ongoing contracts [15]. Moreover, previous Korean studies have demonstrated associations between temporary employment and dietary behaviors such as skipping meals, eating alone, and not using nutritional supplements [16,35].

Concerning the mechanism, temporary employment may induce heightened job stress and insecurity among workers, which in turn can contribute to poor dietary habits among workers. Previous studies have shown that those experiencing high job insecurity are more likely to rely on health risk behaviors such as cigarette smoking and alcohol misuse as coping strategies to reduce stress [7,13,14]. Similarly, job insecurity among temporary employees may make them less likely to prepare or consume healthy meals, and individuals may opt to skip meals or consume junk food [15,16,35]. Additionally, high job demands and low resources are core components of job stress experienced by temporary employees, which may shape the dietary habits of workers [36]. However, the existing literature provides limited insight into the connection between job insecurity or temporary employment and the dietary habits of workers. Further in-depth studies are necessary to clarify the mechanisms by which specific factors related to temporary employment influence the poor dietary habits of workers.

One of the novel findings of our study is that the association between temporary employment and dietary quality is more pronounced among male workers. The observed sex interaction was marginally significant for fixed-term employment and significant for daily employment. While the exact mechanism remains largely unknown, one possible explanation is that because of the predominant role of men as breadwinners, job insecurity resulting from temporary employment may be higher among middle-aged male workers. Indeed, these tendencies are particularly notable in East Asian regions, including South Korea, where the profound influence of Confucian cultural norms underscores the significance of traditional sex-role identity [37]. Similarly, a previous study showed that the effect of temporary employment on health-risk behaviors such as cigarette smoking can be more pronounced among male workers than female workers [14]. Moreover, in the additional analysis, we found that when women take on the role of breadwinner, temporary employment is associated with poorer dietary quality.

Given that poor dietary quality is a significant contributor to the development of cardiovascular risk factors and diseases [38], our findings suggest that the lower dietary quality observed among temporary employees may contribute to the increased risk of cardiovascular abnormalities and mortality, as documented in previous studies [5,6,7,8,9]. Therefore, organizational initiatives aimed at enhancing job stability or implementing workplace programs to improve dietary quality are crucial for promoting healthy eating behaviors [39].

Our study has several limitations. First, due to the cross-sectional nature of our study design, we could not assert a causal effect of temporary employment on dietary quality. Therefore, future prospective studies should be conducted to explore changes in dietary quality following temporary employment. Second, measurement errors, including recall bias, should be considered. Although the KHEI is an evidence-based measurement tool used to assess individuals’ adherence to Korean dietary guidelines, dietary quality was evaluated using the 24-h recall method, which may not fully reflect the usual dietary patterns of workers. In contrast to other surveys that assess dietary quality over two or more days, such as the Healthy Eating Index in the National Health and Nutrition Examination Survey in the United States [40], the KHEI in this study measured dietary intake on a single, randomly selected day of the week. This is one of the key limitations of our dietary quality index. To overcome this limitation, future studies should consider collecting dietary information across multiple days or incorporating a food frequency questionnaire into the index calculation to ensure more accurate measurements. Third, owing to the lack of available information, we could not consider other working conditions that could have influenced the dietary quality of workers. For example, high levels of physical or mental job strain, as well as the work environment, whether at home, in a factory, or an office, can notably influence dietary habits and the availability of time, food, and cooking utensils for meal preparation [41,42]. Fourth, while we examined gender differences in the relationship between temporary employment and dietary quality, participants were classified as either male or female based on their biological characteristics. Although the prevalence of transgender or intersex individuals in Korea is relatively low and most are under 30 years old [43], future data collection in the KNHANES should aim for more inclusive criteria. Fifth, we could not establish a clear causal relationship between employment status and dietary quality due to the interdependence between socio-economic variables, such as educational level, income, occupation type, or other unmeasured confounders, and employment status. Consequently, future studies should consider using alternative designs, such as interventional or quasi-experimental designs, to enable causal interpretation.

Despite these limitations, this study used a nationally representative sample of middle-aged Korean workers, thereby enhancing the generalizability of our findings. Additionally, the examination of differences in sex in the relationship between temporary employment and dietary quality addresses a topic that has been underexplored in the existing literature.

## 5. Conclusions

This study showed that temporary employment was associated with poor dietary quality among middle-aged Korean workers, and this association was more pronounced among male workers. Our study suggests that employment temporariness is a social determinant of dietary quality among workers. Therefore, active policy efforts are required to improve the dietary quality of temporary employees. 

## Figures and Tables

**Figure 1 nutrients-16-01482-f001:**
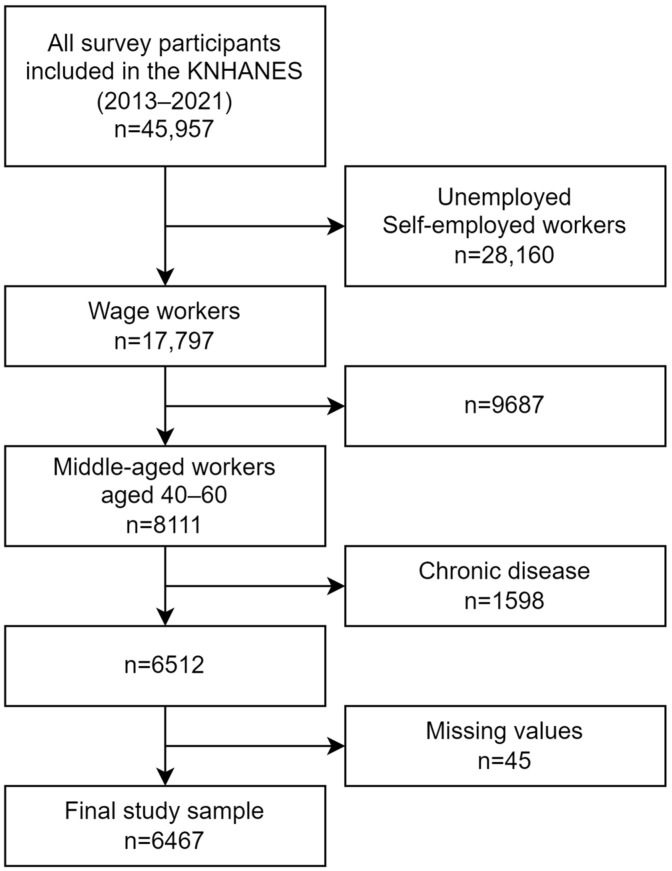
Study sample selection process.

**Table 1 nutrients-16-01482-t001:** Distribution of characteristics according to employment types.

	Overall	Employment Type		
		Regular	Fixed-Term	Daily
	N = 6467	N = 4964	N = 1040	N = 463
**Sex**				
Male	2930 (55.9)	2521 (61.5)	191 (24.2)	218 (57.5)
Female	3537 (44.1)	2443 (38.5)	849 (75.8)	245 (42.5)
**Age**				
40–44	1959 (30.8)	1598 (32.4)	272 (26.8)	89 (20.9)
45–49	1788 (29.2)	1421 (30.1)	270 (27.1)	97 (22.5)
50–54	1362 (21.8)	999 (20.9)	233 (22.9)	130 (30.1)
55–60	1358 (18.2)	946 (16.6)	265 (23.2)	147 (26.5)
**Educational level**				
Middle school or below	845 (11.7)	437 (8.0)	217 (20.3)	191 (37.4)
High school	2480 (37.9)	1791 (35.5)	488 (46.6)	201 (46.9)
College or above	3142 (50.4)	2736 (56.5)	335 (33.1)	71 (15.7)
**Income level**				
Q1	1347 (20.2)	792 (15.5)	343 (33.1)	212 (47.3)
Q2	1623 (24.8)	1186 (23.8)	296 (28.1)	141 (29.6)
Q3	1708 (27.3)	1399 (29.1)	234 (23.3)	75 (14.9)
Q4	1789 (27.7)	1587 (31.5)	167 (15.5)	35 (8.2)
**Marital status**				
Married	5440 (85.1)	4298 (87.6)	816 (78.6)	326 (69.3)
Unmarried	339 (5.7)	234 (5.1)	57 (6.2)	48 (12.1)
Others	688 (9.2)	432 (7.3)	167 (15.2)	89 (18.6)
**Occupational type**				
White collar	3165 (49.4)	2836 (56.7)	298 (29.6)	31 (5.9)
Service or sales workers	1143 (15.9)	720 (13.1)	340 (31.6)	83 (15.4)
Blue collar	2159 (34.7)	1408 (30.2)	402 (38.8)	349 (78.7)
**Weekly working hours**				
<35 h	1469 (20.1)	703 (12.2)	526 (49.9)	240 (49.2)
35–52 h	4149 (65.6)	3601 (73.4)	390 (36.7)	158 (35.6)
≥53 h	849 (14.3)	660 (14.4)	124 (13.4)	65 (15.3)

Values are presented as n (%). The survey weights were adjusted to calculate the proportions.

**Table 2 nutrients-16-01482-t002:** Characteristics of the Korean Health Eating Index according to employment types. Values are presented as weighted mean ± standard deviation.

	Scoring System			Overall Sample	Employment Type		
	Range	Criteria			Regular	Fixed-Term	Daily
		Minimum Score	Maximum Score	N = 6467	N = 4964	N = 1040	N = 463
**Total KHEI score**	0–100			63.0 ± 12.7	63.2 ± 12.6	63.4 ± 12.7	60.3 ± 12.5
**Adequacy component**							
**Total adequacy score**	0–55			31.5 ± 10.0	31.8 ± 9.9	31.2 ± 9.9	28.8 ± 10.7
Have breakfast	0–10	0 times/week	≥5 times/week	7.1 ± 3.9	7.1 ± 3.9	7.0 ± 3.9	7.2 ± 3.9
Mixed grains intake	0–5	0 serving/day	≥0.3 serving/day	2.0 ± 2.1	2.0 ± 2.1	2.1 ± 2.1	1.8 ± 2.1
Total fruits intake	0–5	0 serving/day	≥3/2 (men/women) serving/day	2.3 ± 2.1	2.2 ± 2.1	2.5 ± 2.2	1.9 ± 2.2
Fresh fruits intake	0–5	0 serving/day	≥1 serving/day	2.5 ± 2.3	2.5 ± 2.3	2.8 ± 2.4	2.1 ± 2.4
Total vegetable intake	0–5	0 serving/day	≥8 serving/day	3.7 ± 1.4	3.8 ± 1.4	3.5 ± 1.4	3.6 ± 1.5
Vegetables intake excluding Kimchi and pickled vegetables intake	0–5	0 serving/day	≥5 serving/day	3.4 ± 1.5	3.5 ± 1.5	3.3 ± 1.6	3.1 ± 1.7
Meat, fish, eggs, and beans intake	0–10	0 serving/day	≥5 serving/day	7.4 ± 2.9	7.5 ± 2.8	7.0 ± 3.0	6.5 ± 3.3
Milk and milk product intake	0–10	0 serving/day	≥1 serving/day	3.2 ± 4.4	3.2 ± 4.4	3.0 ± 4.3	2.5 ± 4.2
**Moderation component**							
**Total moderation score**	0–30			21.7 ± 5.9	21.5 ± 5.9	22.8 ± 5.8	22.3 ± 5.9
Percentage of energy from saturated fatty acid	0–10	>10% of total energy intake	<7% of total energy intake	7.5 ± 3.9	7.4 ± 3.9	7.8 ± 3.7	8.0 ± 3.6
Sodium intake	0–10	>6500 mg/day	≤2000 mg/day	6.0 ± 3.4	5.9 ± 3.4	6.8 ± 3.2	6.0 ± 3.6
Percentage of energy from sweets, beverages, and alcoholic drinks ^a^	0–10	>20% of total energy intake	≤10% of total energy intake	8.2 ± 3.2	8.2 ± 3.2	8.2 ± 3.3	8.4 ± 3.1
**Energy balance component**							
**Total energy balance score**	0–15			9.8 ± 4.5	9.9 ± 4.4	9.4 ± 4.5	9.2 ± 4.6
Percentage of energy intake from carbohydrate	0–5	<50% or >75% of total energy intake	55–65% of total energy intake	2.8 ± 2.1	2.9 ± 2.1	2.6 ± 2.1	2.6 ± 2.1
Percentage of energy intake from fat	0–5	<10% or >35% of total energy intake	15–30% of total energy intake	3.7 ± 2.0	3.7 ± 1.9	3.6 ± 2.0	3.6 ± 1.9
Energy intake	0–5	<50% or >75% of the EER	75–125% of the EER	3.3 ± 2.2	3.3 ± 2.2	3.2 ± 2.2	2.9 ± 2.3

KHEI: Korean Healthy Eating Index; ^a^ In the eighth wave of the KNHANES (2019–2021), this item was modified to evaluate the percentage of energy intake from total sugar, while maintaining consistency with the previous standards (sixth and seventh waves) in terms of minimum and maximum score criteria and scoring method.

**Table 3 nutrients-16-01482-t003:** Association between temporary employment and the Korean Healthy Eating Index.

	Model 1	Model 2	Model 3
	*β* (95% CI)	*β* (95% CI)	*β* (95% CI)
**Employment type**			
Regular	0.00 (0.00, 0.00)	0.00 (0.00, 0.00)	0.00 (0.00, 0.00)
Fixed term	0.21 (−0.75, 1.16)	−1.07 (−2.11, −0.04)	−0.45 (−1.57, 0.68)
Daily	−2.92 (−4.34, −1.50)	−2.46 (−3.89, −1.03)	−0.74 (−2.54, 1.06)
**Interaction terms**			
Fixed term × men			−1.88 (−4.02, 0.27)
Daily × men			−3.01 (−5.55, −0.48)

*β*, beta coefficient; CI, confidence interval; Model 1: unadjusted model; Model 2: Model 1 + sex + age + education + income + marital status + occupation types + working hours + survey years; Model 3: Model 2 + interaction terms (temporary employment × sex).

**Table 4 nutrients-16-01482-t004:** Sex-stratified associations between temporary employment and the Korean Healthy Eating Index.

	Men		Women	
	Crude Model	Adjusted Model	Crude Model	Adjusted Model
	*β* (95% CI)	*β* (95% CI)	*β* (95% CI)	*β* (95% CI)
**Employment type**				
Regular	0.00 (0.00, 0.00)	0.00 (0.00, 0.00)	0.00 (0.00, 0.00)	0.00 (0.00, 0.00)
Fixed term	−2.56 (−4.51, −0.60)	−1.69 (−3.71, 0.33)	−0.09 (−1.19, 1.01)	−0.90 (−2.04, 0.25)
Daily	−4.89 (−6.85, −2.93)	−2.60 (−4.63, −0.58)	−0.48 (−2.19, 1.24)	−1.34 (−3.19, 0.52)

*β*, beta coefficient; CI, confidence interval; The adjusted model controlled for age, education, income, marital status, occupation type, working hours, and survey years.

## Data Availability

The raw data of the KNHANES are available to the public, accessible at the KNHANES website (https://knhanes.kdca.go.kr/knhanes, accessed on 23 December 2023).

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
