# Peer review of "Temporary Employment Is Associated with Poor Dietary Quality in Middle-Aged Workers in Korea: A Nationwide Study Based on the Korean Healthy Eating Index, 2013–2021"

_nutrients, 2024, doi:10.3390/nu16101482_

Round 1
Reviewer 1 Report
Comments and Suggestions for Authors
The study provides useful insights into the association between temporary employment and nutritional quality among middle-aged workers in Korea. The results have significant ramifications for public health policy and workplace programs targeted at enhancing dietary habits among temporary employees. The manuscript is well-written/organized.
1. Please remove the “Previous studies have shown that” from abstract section.
2. How the authors have calculated sample size?
Author Response
Thank you for your insightful comments. Please find the attached file.
Reviewer #1.
The study provides useful insights into the association between temporary employment and nutritional quality among middle-aged workers in Korea. The results have significant ramifications for public health policy and workplace programs targeted at enhancing dietary habits among temporary employees. The manuscript is well-written/organized.
Authors’ response) We sincerely thank you for taking the time to read and review our manuscript. Your suggestions were very helpful in improving our work. Please find below our point-by-point response to each of your concern. The revisions and additions made in response to your comments are highlighted in the manuscript. Thank you.
- Please remove the “Previous studies have shown that” from abstract section.
Response 1) Thank you. We have removed the phrase from the abstract.
(Original manuscript)
“Previous studies have shown that temporary employment is associated with an elevated risk of cardiovascular diseases and mortality.” →
(Revised manuscript) Abstract; Line 13-14;
“Temporary employment is associated with an elevated risk of cardiovascular diseases and mortality.”
- How the authors have calculated sample size?
Response 2) Thank you. We did not calculate the sample size because we used a dataset previously established by another party, similar to other studies that use KNHANES data. The KNHANES dataset is already nationally representative and matches our population of interest. Thus, restricting the sample size would offer little advantage. We clarified this information in the methods section.
- Methods; Line 98-101;
“Given that this study involves secondary data analysis, we did not compute a sample size for analysis, as is customary with studies that use KNHANES datasets [1-5]. The KNHANES dataset is constructed to encompass a nationally representative sample of the Korean population, which corresponds to our intended study population.”

Reviewer 2 Report
Comments and Suggestions for Authors
Thank you for submitting the manuscript "Article
Association between Temporary Employment and Dietary
Quality in Middle-Aged Workers in Korea: A Nationwide
Study Based on the Korean Healthy Eating Index, 2013–2021" to Nutrients. Although the manuscript brings to light an interesting subject from the literature, I have important concerns related to the way the experimental part was conducted, especially the ethical aspects of the research that were not reported: voluntary consent from the research participant and prior approval of the project by a research ethics committee Additionally, I have a few other questions:
- Perhaps the authors could change the title to something more attractive, using, for example, the fact that there was an association and this being an interesting result.
- The introduction argues that the type of work influences the KHEA. But what are the reasons for this to happen? Is this related to the fact that salary gains are lower and working hours are longer? You need to provide a little more information on this subject in the introduction.
- It is normal for more than one 24-hour recall to be collected in the same week (two days on weekdays and one day on weekends) so that the actual intake of the research volunteer is considered reliable. Was this accomplished? Which days of the week?
- It seems to me that the issue of sex/gender was a criterion, it was not an inclusive criterion. How was this worked on during data collection?
- Reading the entire manuscript, I had questions about the ethical issues of the work. Was approval from a Research Ethics Committee prior to carrying out the research requested? This Research Ethics Committee approved the execution of the research. An affirmation that Helsinki's declaration has been fulfilled must be provided.
Comments on the Quality of English LanguageModerate editing of English language required.
Author Response
Thank you for your insightful comments. Please find the attached file.
Reviewer #2.
Thank you for submitting the manuscript "Article Association between Temporary Employment and Dietary Quality in Middle-Aged Workers in Korea: A Nationwide Study Based on the Korean Healthy Eating Index, 2013–2021" to Nutrients. Although the manuscript brings to light an interesting subject from the literature, I have important concerns related to the way the experimental part was conducted, especially the ethical aspects of the research that were not reported: voluntary consent from the research participant and prior approval of the project by a research ethics committee Additionally, I have a few other questions:
Authors’ response) We sincerely thank you for taking the time to read and review our manuscript. Your suggestions were very helpful in improving our work. Please find below our point-by-point response to each of your concern. The revisions and additions made in response to your comments are highlighted in the manuscript. Thank you.
Point 1) Perhaps the authors could change the title to something more attractive, using, for example, the fact that there was an association and this being an interesting result.
Response 1) Thank you for the suggestion. We have changed the title as follows:
Title
“Temporary employment is associated with poor dietary quality in middle-aged workers in Korea: A nationwide study based on the Korean Healthy Eating Index, 2013–2021”
Point 2) The introduction argues that the type of work influences the KHEA. But what are the reasons for this to happen? Is this related to the fact that salary gains are lower and working hours are longer? You need to provide a little more information on this subject in the introduction.
Response 2) Thank you. As per your suggestion, we have added more introduction on how type of work can influence the dietary habits of workers.
- Introduction; Line 57-61;
“The relationship between employment type and workers’ dietary habits can be influenced by various complex mechanisms; previous studies have shown that temporary employees are more often subject to factors such as high job stress, long working hours, and lower wages [6,7], which can lead to poorer dietary quality.”
Point 3) It is normal for more than one 24-hour recall to be collected in the same week (two days on weekdays and one day on weekends) so that the actual intake of the research volunteer is considered reliable. Was this accomplished? Which days of the week?
Response 3) Thank you. Unlike other surveys that assess dietary quality over two or more days, the KHEI measures dietary intake on only one randomly selected weekday, which we recognize as a key limitation of our measurement. We have clarified this point in the limitations section.
- Discussion (limitation); Line 260-266;
“In contrast to other surveys that assess dietary quality over two or more days, such as the Healthy Eating Index in the National Health and Nutrition Examination Survey in the United States [8], the KHEI in this study measured dietary intake on a single, randomly selected day of week. This is one of the key limitations of our dietary quality index. To overcome this limitation, future studies should consider collecting dietary information across multiple days or incorporating a food frequency questionnaire into the index calculation to ensure more accurate measurements.”
Point 4) It seems to me that the issue of sex/gender was a criterion, it was not an inclusive criterion. How was this worked on during data collection?
Response 4) Thank you. Participants were classified as either male or female based on their biological characteristics. Although the prevalence of transgender or intersex individuals in Korea is relatively low and most are under 30 years old [9], we recognize that this classification is not inclusive. We have highlighted this limitation in the relevant section.
- Discussion (limitation); Line 271-276;
“Fourth, while we examined gender differences in the relationship between temporary employment and dietary quality, participants were classified as either male or female based on their biological characteristics. Although the prevalence of transgender or intersex individuals in Korea is relatively low and most are under 30 years old [44], future data collection in the KNHANES should aim for more inclusive criteria.”
Point 5) Reading the entire manuscript, I had questions about the ethical issues of the work. Was approval from a Research Ethics Committee prior to carrying out the research requested? This Research Ethics Committee approved the execution of the research. An affirmation that Helsinki's declaration has been fulfilled must be provided.
Response 5) Thank you. Prior to each wave of the KNHANES, approval is obtained from the institutional review board of the KDCA. This secondary data analysis study, which used KNHANES data from 2013-2021, was granted exempt status by the IRB of our affiliated university (IRB number 4-2023-0959) before we conducted our analysis. This is standard procedure for research using KNHANES data. We have also included a reference to the Helsinki Declaration.
- Methods; 2.2. Ethics statement; Line 103-111;
“Before conducting each wave of the KNHANES, ethical approval was obtained from the Institutional Review Board (IRB) of the Korea Disease Control and Prevention Agency (approval numbers: 2013-07CON-03-4C; 2013-12EXP-03-5C; 2018-01-03-P-A; 2018-01-03-C-A; 2018-01-03-2C-A; 2018-01-03-5C-A). Additionally, this secondary data analysis was approved with exempt status by the IRB of Yonsei Health System (IRB number 4-2023-0959) prior to conducting analysis. The authors assert that all procedures contributing to this work comply with the ethical standards of the relevant national and institutional committees on human experimentation and with the Helsinki Declaration of 1975, as revised in 2013.”

Reviewer 3 Report
Comments and Suggestions for Authors
This paper presents a significant quantitative analysis of the relationship between employment status and dietary quality in South Korea. The authors' weighted parameter estimation has yielded intriguing and generally reasonable results, underscoring the importance of this study. As the authors rightly highlight, a significant finding of the study is the stronger association between temporary employment and dietary quality among male workers. However, certain aspects of the paper could benefit from further elucidation.
(1) Daily laborers generally have lower levels of education and lower incomes and are more likely to work in blue-collar jobs. In contrast, full-time workers are highly educated, earn high incomes, and are primarily employed in white-collar jobs (as shown in Table 1). Education level and income level may also be related to dietary quality, education level may affect employment status, and education level and employment status may affect income level. The authors seem to believe that using education and income levels as independent variables eliminates their effects on dietary quality and measures the 'pure effect' of employment status on dietary quality. If so, it is crucial to ensure that there is no correlation between education level, income level, and employment status, as they should be independent. Nevertheless, as Table 1 demonstrates, the assumption of independence between these variables would be easily refuted. This underscores the urgent need for further research to address these potential limitations and ensure the robustness of the findings.
(2) Was the spouses' employment status not included in the survey data? Although it is important to study the correlation between respondent's gender and employment status, as discussed in the paper, considering the common belief that men are the primary earners in East Asia, it is not surprising that the employment status of spouses had a notable effect on the quality of women's diet. However, I think the authors can verify this instead of making assumptions by examining actual data.
Despite the abovementioned questions, the reviewer is confident that minor modifications can effectively address them. These modifications will improve the discussion of the outcome and ensure that the final version meets the desired standards. Therefore, the necessary changes are recommended to achieve the desired quality as an academic paper published in Nutrients.
Author Response
Thank your for your insightful comments. Please find the attached file.
Reviewer #3.
This paper presents a significant quantitative analysis of the relationship between employment status and dietary quality in South Korea. The authors' weighted parameter estimation has yielded intriguing and generally reasonable results, underscoring the importance of this study. As the authors rightly highlight, a significant finding of the study is the stronger association between temporary employment and dietary quality among male workers. However, certain aspects of the paper could benefit from further elucidation.
Authors’ response) We sincerely thank you for taking the time to read and review our manuscript. Your suggestions were very helpful in improving our work. Please find below our point-by-point response to each of your concern. The revisions and additions made in response to your comments are highlighted in the manuscript. Thank you.
(1) Daily laborers generally have lower levels of education and lower incomes and are more likely to work in blue-collar jobs. In contrast, full-time workers are highly educated, earn high incomes, and are primarily employed in white-collar jobs (as shown in Table 1). Education level and income level may also be related to dietary quality, education level may affect employment status, and education level and employment status may affect income level. The authors seem to believe that using education and income levels as independent variables eliminates their effects on dietary quality and measures the 'pure effect' of employment status on dietary quality. If so, it is crucial to ensure that there is no correlation between education level, income level, and employment status, as they should be independent. Nevertheless, as Table 1 demonstrates, the assumption of independence between these variables would be easily refuted. This underscores the urgent need for further research to address these potential limitations and ensure the robustness of the findings.
Response 1) Thank you for the suggestion. As you mentioned, we could not assert the pure, causal effect of employment status on dietary quality due to the interdependence between socio-economic variables and employment status. Therefore, in the limitations section, we emphasized the need for studies using alternative designs, such as intervention studies, that minimize the impact of confounders and allow for causal interpretation.
- Discussion; Line 276-281;
“Fifth, we could not establish a clear causal relationship between employment status and dietary quality due to the interdependence between socio-economic variables, such as educational level, income, occupation type, or other unmeasured confounders, and employment status. Consequently, future studies should consider using alternative designs, such as interventional or quasi-experimental designs, to enable causal interpretation.”
(2) Was the spouses' employment status not included in the survey data? Although it is important to study the correlation between respondent's gender and employment status, as discussed in the paper, considering the common belief that men are the primary earners in East Asia, it is not surprising that the employment status of spouses had a notable effect on the quality of women's diet. However, I think the authors can verify this instead of making assumptions by examining actual data.
Response 2) Thank you for your suggestion. We've carried out an additional analysis on a subsample with data on both the husband's and wife's employment status. The results were intriguing, indicating that women in temporary employment who have a non-working spouse or no spouse had poorer dietary quality. We have described this finding in the discussion section.
- Methods; Line 167-170;
“In our additional analysis, we examined how the relationship between temporary employment and the KHEI differs depending on the employment status of the spouse. We conducted a stratified analysis, comparing groups with a working spouse against those with a non-working spouse or no spouse at all.”
Table S1 Associations between temporary employment and the Korean Healthy Eating Index according to the souse’s employment status.
|
|
Men |
Women |
||
|
|
Working spouse |
Non-working spouse or no spouse |
Working spouse |
Non-working spouse or no spouse |
|
|
β (95% CI) |
β (95% CI) |
β (95% CI) |
β (95% CI) |
|
Employment type |
|
|
|
|
|
Regular |
0.00 (0.00, 0.00) |
0.00 (0.00, 0.00) |
0.00 (0.00, 0.00) |
0.00 (0.00, 0.00) |
|
Fixed-term |
-0.38 (-3.20, 2.45) |
-5.16 (-8.38, -1.95) |
-0.72 (-2.06, 0.62) |
-2.20 (-4.58, 0.19) |
|
Daily |
-3.26 (-6.28, -0.24) |
-3.40 (-6.43, -0.37) |
0.04 (-2.10, 2.19) |
-5.80 (-9.73, -1.86) |
β, beta coefficient; CI, confidence interval
The adjusted model controlled for age, education, income, occupation type, working hours, and survey years.
- Results; Line 199-202;
“Table S1 presents the results of an additional analysis, showing that among women whose spouse is not employed or who do not have a spouse, temporary employment is associated with lower dietary quality compared to regular employment.”
- Discussion; Line 244-245;
“Moreover, in the additional analysis, we found that when women take on the role of breadwinner, temporary employment is associated with poorer dietary quality.”z
Despite the abovementioned questions, the reviewer is confident that minor modifications can effectively address them. These modifications will improve the discussion of the outcome and ensure that the final version meets the desired standards. Therefore, the necessary changes are recommended to achieve the desired quality as an academic paper published in Nutrients.
Response) Thank you for your insightful comments for our work.

Round 2
Reviewer 2 Report
Comments and Suggestions for Authors
This reviewer appreciates the fact that the authors made an effort to make the suggested corrections. The major concerns raised by this review were not addressed as the work had already been outlined. However, at least they were corrected in the text and added as a limitation of the work.